# Analysis of Adverse Drug Reactions in Pediatric Patients with Epilepsy: An Intensive Pharmacovigilance Study

**DOI:** 10.3390/children10111775

**Published:** 2023-11-01

**Authors:** Ernestina Hernández García, Lizbeth Naranjo, Luz Adriana Pichardo-Macías, María Josefa Bernad Bernad, Lucila Isabel Castro-Pastrana, Matilde Ruíz García, Tanya Alejandra García Bernal, Jessica Lizbeth Mendoza Solís, David Calderón Guzmán, Luisa Díaz-García, Julieta Griselda Mendoza-Torreblanca, Juan Luis Chávez Pacheco

**Affiliations:** 1Laboratorio de Farmacología, Subdirección de Medicina Experimental, Instituto Nacional de Pediatría, Ciudad de Mexico 04530, Mexico; ernestina_hg@aol.com; 2Programa de Maestría y Doctorado en Ciencias Médicas, Odontológicas y de la Salud, Facultad de Medicina, Universidad Nacional Autónoma de México (UNAM), Ciudad de Mexico 04960, Mexico; 3Departamento de Matemáticas, Facultad de Ciencias, Universidad Nacional Autónoma de Mexico, Ciudad de Mexico 04510, Mexico; lizbethna@ciencias.unam.mx; 4Departamento de Fisiología, Instituto Politécnico Nacional, Escuela Nacional de Ciencias Biológicas, Ciudad de Mexico 07738, Mexico; lpichardom@ipn.mx; 5Departamento de Farmacia, Facultad de Química, Universidad Nacional Autónoma de Mexico, Ciudad de Mexico 04510, Mexico; bernadf@comunidad.unam.mx; 6Departamento de Ciencias Químico Biológicas, Universidad de las Américas Puebla, Puebla 72810, Mexico; lucila.castro@udlap.mx; 7Servicio de Neurología, Dirección Médica, Instituto Nacional de Pediatría, Ciudad de Mexico 04530, Mexico; matilderuizg@gmail.com; 8Farmacia Intrahospitalaria, Hospital General Dr. Manuel Gea González, Ciudad de Mexico 14080, Mexico; mix_aynat@hotmail.com; 9Facultad de Medicina, Universidad Nacional Autónoma de Mexico, Ciudad de Mexico 04360, Mexico; jmendoza514@gmail.com; 10Laboratorio de Neurociencias, Subdirección de Medicina Experimental, Instituto Nacional de Pediatría, Ciudad de Mexico 04530, Mexico; solodavid2001@yahoo.com.mx (D.C.G.); julietamt14@hotmail.com (J.G.M.-T.); 11Departamento de Metodología de la Investigación, Subdirección de Investigación Clínica, Instituto Nacional de Pediatría, Ciudad de Mexico 04530, Mexico; mdiazg@pediatria.gob.mx

**Keywords:** adverse drug reactions, epilepsy, anti-seizure medication, logistic regression, risk factors, generalized estimating equations, valproic acid, levetiracetam, phenytoin

## Abstract

Epilepsy is a chronic neurological disease characterized by the presence of spontaneous seizures, with a higher incidence in the pediatric population. Anti-seizure medication (ASM) may produce adverse drug reactions (ADRs) with an elevated frequency and a high severity. Thus, the objective of the present study was to analyze, through intensive pharmacovigilance over 112 months, the ADRs produced by valproic acid (VPA), oxcarbazepine (OXC), phenytoin (PHT), and levetiracetam (LEV), among others, administered to monotherapy or polytherapy for Mexican hospitalized pediatric epilepsy patients. A total of 1034 patients were interviewed; 315 met the inclusion criteria, 211 patients presented ADRs, and 104 did not. A total of 548 ASM-ADRs were identified, and VPA, LEV, and PHT were the main culprit drugs. The most frequent ADRs were drowsiness, irritability, and thrombocytopenia, and the main systems affected were hematologic, nervous, and dermatologic. LEV and OXC caused more nonsevere ADRs, and PHT caused more severe ADRs. The risk analysis showed an association between belonging to the younger groups and polytherapy with ADR presence and between polytherapy and malnutrition with severe ADRs. In addition, most of the severe ADRs were preventable, and most of the nonsevere ADRs were nonpreventable.

## 1. Introduction

Epilepsy is a chronic neurological disease with diverse etiology that affects all age groups, with a higher incidence in the child population [1,2]. It is characterized by the presence of spontaneous and recurrent seizures whose clinical manifestations are highly variable and depend on the brain area involved [3,4,5]. Worldwide, ~50 million people have epilepsy, making it one of the most common brain disorders [5,6]. The incidence of epilepsy in developed countries ranges from 42 to 61 per 100,000 inhabitants [5,7] and is usually close to double or more in developing countries [2,5,8]. In Mexico, the estimated prevalence is 1% to 2% in the general population and ~1.8% in the child population [9]. At the National Institute of Pediatrics (INP; Mexico City, Mexico), a third-level hospital, approximately 47% of patients who request a consultation in the Neurology Department suffer from this disease [10].

The appropriate scheme for epilepsy treatment is very important since epilepsy can be disabling and even fatal. For this, it is important to distinguish between types of epilepsy and types of seizures because the proper epilepsy classification allows for the prediction of its prognosis and the assessment of possible etiology; however, treatment options are mainly based on the classification of the types of seizures, focal or generalized, exhibited by the patient [11,12]. Monotherapy has been the gold standard for epilepsy; in approximately 70% of patients with newly diagnosed epilepsy, initial treatment with a single anti-seizure medication (ASM) leads to complete seizure control; nonetheless, when monotherapy fails, in an important minority of patients, the alternative is a second (or multiple)-line agent [13]. Unfortunately, the use of polytherapy significantly increases side effects, complex drug interactions, and costs [14,15].

ASM, like most drugs, are potentially dangerous and produce adverse drug reactions (ADRs), which are defined by the World Health Organization as “Any noxious, unintended and undesired effect of a drug which occurs at the dosages used in humans for prophylaxis, diagnosis or therapy”, and the causal relationship is reasonably attributable to the drug [16,17]. ADRs are a major clinical and economic problem in pediatric medicine, and systematic reviews and meta-analyses have shown that the overall incidence of ADRs is ~10% among hospitalized and 1–1.5% among outpatient children [18,19]. The annual cost of adverse events in hospital care amounts to hundreds of billions of US dollars worldwide [20]. Prospective studies have suggested that between one-third and up to two-thirds of children who receive ASM will experience an ADR and that the risk of an ADR is significantly higher among patients receiving polytherapy than among those receiving monotherapy [15,21,22]. Additionally, ASM-ADRs have been associated with fatality, elevated frequency, and high severity [15,23,24]; however, clinical information regarding their frequency, efficacy, and safety is often not registered and, therefore, not reported.

In this context, intensive pharmacovigilance (IPV) is the systematic monitoring of the occurrence of ADRs resulting from drug use during the entire length of prescription and is considered a useful tool to prevent, identify, and treat preventable and nonpreventable ADRs to medications [17,25]. This methodology is capable of identifying signals for events that were not necessarily suspected to be ADRs of the drug studied and may estimate the incidence of adverse events, enabling the quantification of the risk of certain ADRs [26,27]. IPV activities have been demonstrated to favor the assessment of drug safety [17]; however, few clinical studies have focused on detecting ASM-ADRs, especially those related to hospitalized pediatric patients. Therefore, the objective of the present study was to evaluate, through IPV over 122 months, the ADRs provoked by the ASM with the highest prescription rates in the INP, which were administered in mono- or polytherapy to mexican hospitalized pediatric patients with a diagnosis of epilepsy.

## 2. Materials and Methods

### 2.1. Study Design, Settings, and Ethical Considerations

This was a cohort, prospective, and observational study based on IPV. The protocol was registered and approved on 26 September 2012, by the Institutional Research and Ethics Board (INP registration number: 090/2012; IRB00013674). All procedures were conducted in accordance with the ethical standards of the Institutional Research Committee and the Code of Ethics of the World Medical Association Declaration of Helsinki. This study was carried out in the hospitalization area of neurology from October 2012 to December 2022.

All recruited patients and parents were adequately informed about the objectives, methods, probable benefits, foreseeable risks, and discomforts derived from this study. Additionally, informed consent forms were signed by the father and mother or tutor. Patients over 12 years of age signed the assent form as long as they knew how to read and write and did not present an overall delay in their neurodevelopment.

### 2.2. Participants

Hospitalized pediatric patients with a diagnosis of epilepsy, defined by the type of seizure according to the “ILAE 2017 classification of seizure types basic version” (focal onset, focal onset aware, focal onset impaired awareness, focal onset motor, focal onset nonmotor, focal to bilateral tonic–clonic, generalized onset, generalized onset motor, generalized onset nonmotor and unclassified) [28], with complete files and source documents (laboratory and cabinet studies) and whose main treatment, either in monotherapy or polytherapy, was with any of the following ASM were included: valproic acid (VPA), carbamazepine (CBZ), oxcarbazepine (OXC), phenytoin (PHT), levetiracetam (LEV), and topiramate (TPM). Patients (or their parents) were free to withdraw from this study at any time.

### 2.3. General Procedure

Eligible patients were identified and informed about this study; if they met the inclusion criteria, they were invited to participate and then signed the informed consent/assent form. IPV was implemented for the hospitalized patients, i.e., we carried out systematic noninterventional observational daily visits to closely monitor and detect the presence of ADRs during the hospitalization of the pediatric patients.

### 2.4. Data Collection and Processing

The doctor performed the anamnesis of the patients and registered the information in the clinical record. Then, the data were collected in a specific format designed for this study. The main data obtained from the clinical records were age, sex, seizure type, weight, height, prescribed drugs, and dose at the beginning and end of treatment, as well as the record of the withdrawal of the drug in case the treating physician considered it necessary and the suspicion of the presence of adverse reactions. Patients with antiepileptic treatment were visited daily to identify and register suspected adverse reactions that occurred during the hospital stay and when the drugs were being administered. In the case of laboratory studies and electroencephalograms (EEGs), data were collected during hospitalization and prior to discharge and served to corroborate the diagnosis of epilepsy and the presence of some ADRs.

### 2.5. Instruments

The classification of the age groups and nutritional status was carried out according to the Official Mexican Standard NOM-008-SSA2-1993, “Control of nutrition, growth and development of children and adolescents. Criteria and procedures for the provision of the service” [29]. The Naranjo algorithm was used to evaluate the causality of an ADR as definite, probable, possible, or doubtful [30,31]. The Hartwig and Siegel Scale was used to assess the severity of the ADRs [32,33]. The official Mexican Norm NOM-220-SSA1-2016 was employed to classify the severity as mild, moderate, or severe and the seriousness as severe or nonsevere [34]. Finally, the Schumock and Thornton criteria were used to determine preventability as preventable, probably preventable, or nonpreventable [35,36].

### 2.6. Data Analysis

To analyze the proportions of patients who presented ASM-ADRs, the proportions of severe and nonsevere ADRs and the variables of causality, severity, seriousness, preventability, frequencies, and relative frequencies were computed. The association between two categorical variables was examined using the odds ratio (OR) and Pearson’s chi-squared test [37]. To analyze the presence vs. absence of ASM-ADRs and their associated risk factors, a logistic regression model was used, first by evaluating each variable separately in a univariate logistic regression model, and then including all variables together in a multivariate logistic regression model [37,38]. To analyze the seriousness of ASM-ADRs (severe vs. nonsevere) and their associated risk factors, the generalized estimating equation (GEE) model was used, first evaluating each variable separately in a univariate GEE model, and then including all variables together in a multivariate GEE model [39,40]. The GEE model accounted for dependency in repeated measures (some children presented more than one ASM-ADR). In both univariate and multivariate models, to determine the risk factors associated with the response variable, the OR with 95% confidence intervals (95% CIs) and their associated *p* values were estimated. Statistical analysis was performed using R version 4.1.1. and RStudio version 1.4.1717 software, using the packages “stats”, “ggplot2”, [41] and “geepack” [42]. We considered a *p* value < 0.05 to be statistically significant.

## 3. Results

### 3.1. Characteristics of Seizures; the Presence of ASM-ADRs and EEGs of Patients with Epilepsy

A total of 1034 pediatric patients with a diagnosis of epilepsy were surveyed. A total of 315 patients met the inclusion criteria; of these, 211 patients had ADRs, and 104 did not. Regarding their EEGs, 228 patients had an abnormal EEG, 17 had a normal EEG, and 70 patients did not undergo an EEG. Table 1 shows the frequencies and distribution of seizure type, presence of ADRs, and EEG characteristics of the patients.

### 3.2. Risk Factors Associated with the Presence of ADRs

The results of the univariate and multivariate logistic regression models are shown in Table 2. The second and third columns display the frequencies and distribution of patients who presented and did not present ASM-ADRs by sex, age (at study entry), seizure type, therapy, and nutritional status. The fourth and fifth columns display the results from the univariate logistic regression, and the sixth and seventh columns exhibit the results from the multivariate logistic regression.

Regarding sex, 176 boys (55.9%) and 139 girls (44.1%) were included in this study. No significant differences in the presence of ASM-ADRs by sex were found in either the univariate or multivariate analysis. With respect to age range, the most frequent population comprised infants, with 82 patients (26.0%). The ORs for the presence of ASM-ADRs for infants, older infants, preschool children, and school-aged children were 2.80, 1.91, 2.69, and 2.28 times higher than those for adolescents, respectively. There was a significant difference in infant, preschool, and school-age children with respect to adolescent age in the univariate and multivariate analyses. Thus, the risk analysis indicated that belonging to the younger groups significantly increased the probability of presenting ADRs.

According to seizure type, in univariate and multivariate analyses, there were no significant differences between generalized and focal seizures. With respect to therapy, the results showed that the presence of ASM-ADRs for polytherapy was 2.47 and 2.08 times higher than that for monotherapy in both univariate and multivariate analyses, respectively. This indicated a greater probability of presenting ADRs among patients treated with more than one ASM. Finally, there was no significant difference in the presence of ASM-ADRs in terms of nutritional status.

### 3.3. Frequency and Incidence of ASM-ADRs

Among the 211 patients who presented ADRs, a total of 548 ASM-ADRs were identified. In total, 80 patients presented at least 1 ADR, 53 had 2 ADRs, and 34 had 3 ADRs, with a total of 167 patients who presented with 1–3 ADRs, with 288 ADRs and an average of 1.7 ADRs per patient. Among the remaining patients (44), 4 to 14 ADRs were observed, with a total of 260 ADRs and an average of 5.9 ADRs per patient. Note that a small number of patients presented a large number of ADRs (from 4 to 14 ADRs; Figure 1).

### 3.4. Type, Frequency, and Distribution of the Main ASM-ADRs by Drug

The most frequent ASM-ADRs observed were drowsiness, irritability, thrombocytopenia, alopecia, and low VPA levels; this corresponded to 47.6% of the total ADRs detected (Table 3). Other clinically important reactions were hyperammonemia, erythema, constipation, low PHT levels, and high VPA levels; the sum of all the aforementioned ADRs reached 68.1%. Seventy-eight percent of ADRs were reached when events such as neutropenia, uncontrolled seizures, elevated hepatic enzymes, metabolic acidosis, rash, and supratherapeutic PHT levels were added. Other ADRs with lower frequency were edema, liver damage, etc. (Table 3).

Additionally, Table 3 shows the frequency and distribution of the ASM-ADRs caused by VPA, LEV, PHT, OXC, and other drugs. Patients treated with VPA, LEV, and PHT had the highest number of ADRs observed (480; 87.6% overall). It is important to mention that the concomitant prescription of VPA and PHT was identified in 20 patients. According to the literature, there is a pharmacological interaction of these two drugs [43], which could explain the observation of an increase in plasma levels of PHT and a decrease in VPA levels simultaneously in 5 patients. Other observed interactions were VPA-LEV (low and supratherapeutic VPA levels) and PHT-OXC (supratherapeutic PHT levels).

### 3.5. Organs and Systems Affected by Drugs

In addition, Table 4 shows the organs and systems affected by the ASM-ADRs.

The hematologic system was affected with 189 ADRs (34.5%), the nervous system with 183 ADRs (33.4.0%), the dermatologic system with 105 ADRs (19.2%), and the gastrointestinal system with 54 ADRs (9.8%). The cardiovascular, immunological, respiratory, and musculoskeletal systems were minimally affected (Table 4). Also shown are the frequency and distribution of the 548 ADRs that affected organs and systems associated with the following ASM: VPA, LEV, PHT, OXC, TPM, and clonazepam (CZP). Other ASM that were administered concomitantly and for which ADRs could be identified were CBZ, clobazam (CLB), and gabapentin (GBP). It should be mentioned that a severe ADR (Stevens–Johnson syndrome) caused by the administration of OXC to a patient was detected.

### 3.6. Causality Assessment of ASM-ADRs

After employing the modified Naranjo algorithm, it was found that of 548 ADRs, 100 (18.3%) were definite, 351 (64.0%) were probable, 89 (16.2%) were possible, and 8 (1.5%) were doubtful. Figure 2 shows the ADR causality classification by drug. VPA caused 55 definite ADRs, 167 probable, 28 possible, and 3 doubtful; LEV caused 8 definite ADRs, 86 probable, 33 possible, and 4 doubtful; PHT caused 19 definite ADRs, 57 probable and 20 possible; OXC caused 7 definite ADRs, 15 probable and 5 possible; TPM caused 5 definite ADRs, 10 probable and 2 possible; CZP caused 1 definite ADR, 9 probable and 1 doubtful; CBZ caused 3 definite ADRs, 5 probable and 1 possible; CLB caused 2 definite ADRs; and GBP caused 2 probable ADRs. For statistical analysis, since there were very few doubtful ADRs, they were grouped into the “possible” group. Note that most ADRs were classified as probable. The results showed a significant difference among the different causality degrees with VPA, LEV, PHT, OXC, and CZP. No significant difference was observed in the degree of causality with TPM, CBZ, CLB, or GBP.

### 3.7. Severity Assessment of ASM-ADRs

The Hartwig and Siegel scale was employed to assess the severity of ADRs. Table 5 shows the classification of severity levels of the clinical manifestations of the ASM-ADRs. A total of 91.3% of the ADRs were classified as levels 1–3, and 8.7% of the ADRs were classified as levels 4–7. This information allowed for us to understand the management of the treatment or procedure needed, as well as the associated damage to pediatric patients with epilepsy.

### 3.8. Severity Assessment of ASM-ADRs According to NOM-220-SSA1-2016

Another way that the severity of ADRs was evaluated was with the Official Mexican Norm NOM-220-SSA1-2016. Figure 3 shows the ADR severity. 

When dividing the severity of ADRs generated by each drug, it was observed that most of them caused mild ADRs: VPA (117, 46%), LEV (98, 75%), PHT (24, 25%), OXC (16, 59%), and TPM (7, 41%). All ADRs of CZP (11) and GBP (2) were mild. Severe ADRs were frequent for drugs such as VPA (106, 42%), LEV (22, 17%), PHT (55, 57%), OXC (6, 22%), and TPM (10, 59%). Statistical analysis showed a significant difference among the severity degrees of ADRs (mild, moderate, and severe) caused by VPA, LEV, PHT, OXC, TPM, and CZP. No significant difference was observed in the degree of severity of the ADRs caused by CBZ, CLB, or GBP. It is important to point out that LEV, OXC, and CZP caused milder ADRs, and PHT and TPM caused more severe ADRs.

### 3.9. Seriousness Assessment of ASM-ADRs According to NOM-220-SSA1-2016

By dividing the ADR seriousness into severe and nonsevere (Figure 4), it was observed that drugs such as LEV (101, 77%) and OXC (16, 59%) caused mainly nonsevere ADRs. All ADRs of CZP (11, 100%) and GBP (2, 100%) were nonsevere. PHT generated mostly severe ADRs (72, 75%). VPA and CLB produced almost equally severe or nonsevere ADRs (VPA 135, 53% severe and 118, 47% nonsevere; CBZ 5, 56% severe and 4, 44% nonsevere). Statistical analysis showed a significant difference among the seriousness degrees of ADRs caused by LEV, PHT, and CZP. No significant difference was observed in the degree of seriousness of the ADRs caused by VPA, OXC, TPM, CBZ, CLB, or GBP.

### 3.10. Risk Factors Associated with the Seriousness of ADRs

Table 6 shows the results of the risk factors associated with seriousness; the second and third columns display the frequencies and distribution of patients who presented severe and non-severe ASM-ADRs by sex, age, seizure type, therapy, and nutritional status. The fourth and fifth columns display the results from the univariate GEE model, and the sixth and seventh columns exhibit the results from the multivariate GEE model. The results showed that therapy and nutritional status were risk factors for severe ADRs. For polytherapy, the ORs were 1.84 and 2.11 times higher than those for monotherapy in the univariate and multivariate analyses, respectively, indicating a greater probability of presenting severe ADRs among patients treated with more than one ASM. With respect to nutritional status, severe malnutrition and mild malnutrition had ORs of 1.67 and 2.06 times higher than that for normal weight in the multivariate analysis.

### 3.11. Preventability Assessment of ASM-ADRs

Based on the analysis with the Shumock and Thornton algorithm, a total of 144 ADRs (26.3%) were preventable, 340 (62.0%) were probably preventable, and 64 (11.7%) were nonpreventable. The main causes for which preventable ADRs occurred were (1) the administration of an inappropriate medication for the clinical condition of the patient and (2) a history of allergies or previous adverse reactions. Likewise, the main causes for which the probably preventable ADRs occurred were (1) that preventive measures were not applied according to the conditions or pathology of the patient, (2) that there was no periodic monitoring of the plasmatic levels of the ASM, and (3) that the available information was not consulted or used to evaluate the presence of drug interactions of the ASM when they were administered concomitantly.

To analyze the association between preventability and severity and between preventability and the seriousness of the ADRs, we used Pearson’s chi-squared test. Table 7 shows that preventable ADRs were associated with severe ADRs, and nonpreventable ADRs were associated with mild ADRs. Table 8 shows that preventable ADRs were associated with severe ADRs and that nonpreventable ADRs were associated with non-severe ADRs.

## 4. Discussion

The most important findings in this study were that 315 patients were enrolled, 211 presented ADRs, and 104 did not. A total of 548 ADRs were identified. Drowsiness, irritability, thrombocytopenia, and alopecia were the most frequent. VPA, LEV, and PHT were the drugs most frequently associated with ADRs; this could be attributed to the fact that these three drugs are the most prescribed at the INP due to their cost—benefit ratio. In addition, the hematologic, nervous, dermatologic, and gastrointestinal systems were the most affected. The logistic regression analysis showed that both age and polytherapy were risk factors associated with the presence of ADRs. Likewise, most of the ADRs were classified as definite or probable. For severity, 48.2% of ADRs were severe, and 51.8% were non-severe. VPA caused similar percentages of severe vs. non-severe ADRs; however, LEV provoked more non-severe ADRs, and PHT provoked more severe ADRs. The GEE multivariate analysis showed that both polytherapy and malnutrition were risk factors associated with the severity of ADRs. Finally, the results showed that most of the severe ADRs were preventable, and most of the non-severe ADRs were nonpreventable.

Very often, the main treatment for pediatric epilepsy is the use of ASM that helps control seizures; similar to other drugs, pharmacological treatment can cause ADRs in patients receiving them [44]. In the present study, the most commonly used ASM was VPA, followed by LEV and PHT. These findings were partially in agreement with those of Kaushik, who reported that the most common drug was VPA, followed by PHT, OXC, LEV, and CLB in Indian pediatric patients [45]. Similarly, Anderson reported the use of VPA followed by OXC, lamotrigine, and LEV in a study conducted in the United Kingdom [15]. In contrast, in a cross-sectional study in Indian children, George et al. found that CLB was used most frequently, followed by PHT, LEV, OXC, and CBZ [46]. The fact that an older/conventional drug, VPA, was used as a first-line drug is due to its broad spectrum of action that can be used to treat both focal and generalized seizures; in addition, it is relatively less expensive than newer ASM and is easily available in worldwide hospital pharmacies.

Our results show that patients with polytherapy had more ADRs than patients with monotherapy; similarly, Anderson et al. showed that a higher percentage of patients on polytherapy experienced ADRs compared with patients on monotherapy [15]. In contrast, no significant difference in ADRs between monotherapy and polytherapy was reported by Bansal and colleagues [21]. In addition, our univariate and multivariate logistic regression analyses showed that both age and polytherapy were risk factors associated with the presence of ADRs. In particular, younger patients (1 m and <1 yr) had the highest probability of presenting ADRs (74.4%). Similarly, in the Priyadharsini study, nearly 60% of the ADRs occurred in patients less than 1 year of age [47]. Belonging to younger groups and/or being treated with two or more ASM significantly increases the probability of presenting ADRs.

VPA is widely used as an ASM that is highly effective in both children and adults. It is one of the first-line drugs for the treatment of epilepsy. The most frequent ADRs include somnolence, weight gain, fatigue, and headache [48,49]. The most serious ADRs include hepatotoxicity and pancreatitis, both of which can result in death [50]. In the present study, somnolence, thrombocytopenia, and irritability were the principal ADRs found with VPA treatment. Previous investigations have shown an increase in sleep need during VPA treatment [51,52]. VPA can increase the brain content of GABA by both stimulating GABA synthesis (by glutamate decarboxylase) and inhibiting GABA degradation (by GABA transaminase and succinic semialdehyde dehydrogenase) [48], which can result in the inhibition of the central nervous system (CNS) and produce drowsiness in some patients. Thrombocytopenia is the most common adverse hematologic effect of VPA, with an incidence varying from 5% to 60% [53,54,55]. The exact mechanism(s) inducing thrombocytopenia has not yet been elucidated, but the immune-mediated destruction of platelets and direct toxicity to the bone marrow have both been hypothesized as possible etiologies [56]. Some data from pediatric patients with epilepsy treated with VPA have indicated behavioral alterations (irritability, hyperactivity, and aggressiveness) [49], similar to our results. Despite these reports, it is not known why VPA causes irritability in children; in fact, it is currently used as a mood stabilizer [57].

LEV is a broad-spectrum ASM with a unique mechanism of action that is able to modulate neurotransmission release by interacting with SV2A [58]. Compared to older ASM, LEV has a good tolerability profile and minimal drug—drug interactions, making it a good option for use in pediatric patients [59]. There have been reports of minor serious short- and long-term ADRs and complications. In a retrospective study of 231 consecutive pediatric patients, the most reported side effects were irritability, hyperactivity, somnolence, behavioral disorders, restlessness, increased seizure frequency, enuresis, headache, and attempted suicide [60]. Our data indicate that behavioral problems and somnolence are the most common adverse events to LEV, which is in accordance with the literature [61,62,63]. The exact mechanism(s) inducing somnolence and behavioral side effects have not yet been elucidated. Somnolence may reflect the depressant CNS effect of LEV, which is necessary to decrease epileptic activity. In particular, this effect has been associated with the predominant effect of LEV on enhanced GABAergic activity [64,65]. The development of negative behavioral symptoms has been associated with inhibiting the action of N-type calcium channels [66].

PHT has been used in the management of epilepsy for over half a century. According to international management guidelines, PHT is the second-line drug used for pediatric convulsive status epilepticus [67,68]. In a prospective study in which 22 children received a total of 100 doses of PHT over a 10-month period, 6 patients presented ADRs, including extravasation of the drug, hypotension, and cardiac arrhythmia. No patient developed skin necrosis, including “purple glove syndrome”, which is commonly associated with the use of PHT [69], but in our study, the mentioned syndrome was not observed. The most common ADR related to PHT was a decrease in drug levels. It is important to mention that some patients used concomitant administration of VPA and PHT. Therapeutic drug monitoring based on the total concentration of PHT may be misleading when VPA is coadministered. The mechanisms involved in the VPA-PHT interaction show that VPA displaces PHT on the plasma protein, thereby enhancing the systemic clearance of the drug and resulting in a decrease in the total drug concentration [70,71].

In the present study, a common ADR reported was mild alopecia, and some ASM precipitate hair loss by inducing the premature rest of follicles (telogen effluvium). Although the mechanism by which ASM provokes alopecia remains unclear, there are reports linked with abnormal concentrations of zinc [72]. Therefore, drugs that decrease zinc concentrations, such as LEV [73,74] and VPA [75], cause alopecia. LEV enhances GABAergic transmission in contrast to zinc antagonism at GABA_A_ and glycine receptors [76] and VPA-induced zinc chelation [75]. The literature indicates that they occur in a dose-dependent manner and may resolve spontaneously despite the continuation of treatment.

According to the modified Naranjo algorithm, most of the ADRs were classified as definite or probable, i.e., in most cases, the related drug effectively caused the ADRs, which is analogous to the findings of a study performed by Kaushik, in which the relationship between the ADRs and the respective drugs was found to be “probable” in 91.3% of cases, followed by possible in 8.7% of cases [45].

As mentioned above, 48.2% of the adverse reactions were serious, and 51.8% were nonserious. Almost the same percentage of serious (53%) and nonserious (47%) ADRs were caused by VPA. However, LEV provoked more non-severe ADRs (77%), and PHT provoked more severe ADRs (75%). It is important to mention that although VPA had the same proportion of severe and non-severe ADRs, the literature indicates that the severe ADRs presented may result in fatality [50]. Moreover, it has been reported that when the drug is indicated in polytherapy, fatalities are significantly more frequently reported than nonfatalities and appear to remain a considerable risk factor for serious ADRs, including hepatotoxicity [50]. In the case of LEV, it is necessary to emphasize that although our study indicated that there was a lower number of severe adverse reactions, the evidence from observational studies shows an increase in behavioral deterioration following LEV treatment [61,62,63].

In this context, our GEE multivariate analysis showed that both polytherapy and malnutrition were risk factors associated with severe ADRs. Nutritional status is a very important aspect to consider since numerous papers have dealt with the effect of ASM on weight and the possible adverse effects of malnutrition in the onset of seizures [77,78]. Our data suggest that children with severe and moderate malnutrition have a greater probability of presenting an ADR than children with obesity. This may be because malnourished children have a decreased hepatic enzyme activity that results in alterations in drug metabolism and pharmacokinetics (accumulation and toxic drug effects) [79]. However, although we did not observe that obesity was a determining factor for ADRs in children, it is a common comorbidity for pediatric epilepsy patients related to worse evolution of the disease [80,81].

Additionally, the results showed that most of the severe ADRs were preventable, while the non-severe ADRs were nonpreventable. Similarly, Mistry and Kaushik reported that the majority of ADRs were preventable [82]. This is an important aspect because ADRs in the pediatric population are a major public health problem that, despite efforts to reduce the incidence of medication-related adverse events, morbidity and mortality, continue to be unacceptably high. The fact that a significant percentage of ADRs could be preventable is a call to the public health service, which is often exceeded in many aspects.

Finally, it is important to point out several limitations of this study. For example, in the statistical analysis, we did not consider the etiology of the epilepsy, whether the patients had refractory epilepsy or comorbidities, or the interaction with other non-anti-seizure medication. However, all these variables will be analyzed in greater detail in future reports. Moreover, observational studies per se have their own limitations, such as the lack of no control over the assignation of treatment to subjects to guarantee balanced samples, mainly under the variables sex, age, geographic area, and, perhaps, under seizure type and nutritional status, among others, as there are in clinical studies (clinical trials). However, they may be the only way to determine the temporal sequence between the exposure variable and the outcome variable, in addition to allowing different outcome variables to be studied simultaneously [83].

## 5. Conclusions

In this study, we observed 548 ADRs in 211 hospitalized pediatric patients with epilepsy. VPA, LEV, and PHT were the main related drugs. Polytherapy was the main risk factor for the presence and severity of these ADRs. In addition, age was a risk factor for the presence of ADRs, indicating that belonging to younger groups significantly increases the probability of having ADRs. Furthermore, malnutrition was the other risk factor for severe ADRs, which can be attributable to alterations in the drug metabolism and pharmacokinetics of these patients. LEV and OXC caused more non-severe ADRs, and PHT caused more severe ADRs. Finally, we showed that most of the severe ADRs were preventable and that most of the non-severe ADRs were nonpreventable. Since the INP is a referral, government-funded, teaching hospital that cares for pediatric patients with third-level pathologies coming from all over the country, conducting intensive pharmacovigilance studies such as ours is essential to begin to understand the safety and efficacy of medications in pediatric patients.

## Figures and Tables

**Figure 1 children-10-01775-f001:**
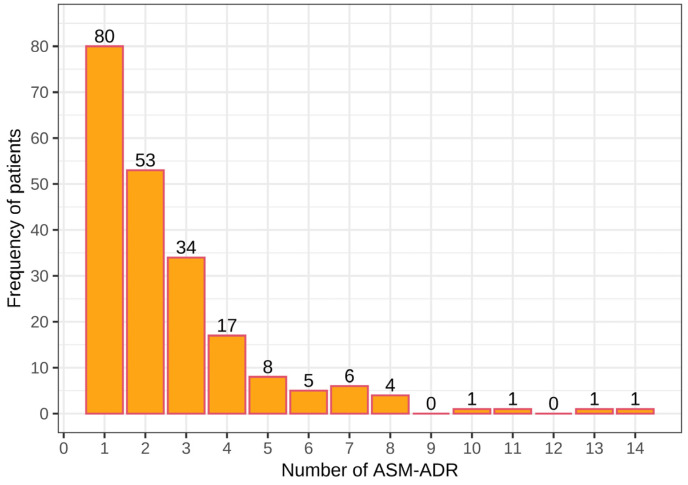
Number of adverse drug reactions due to antiepileptics (ASM-ADR) among pediatric patients with epilepsy.

**Figure 2 children-10-01775-f002:**
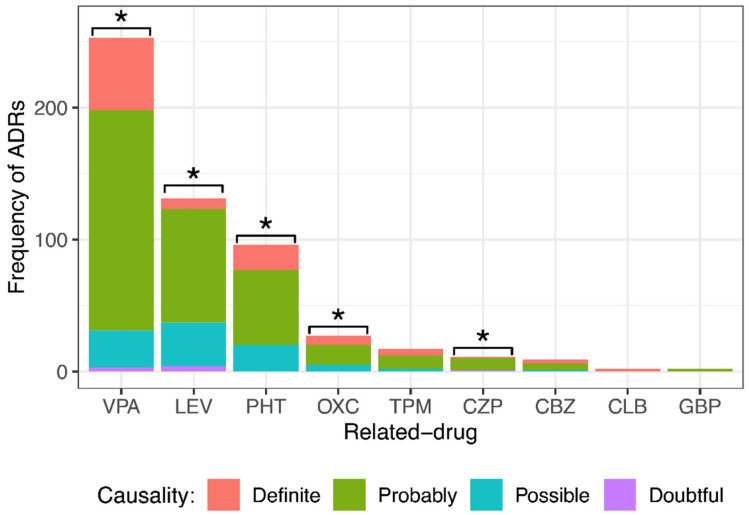
Comparison of different causality degrees for each anti-seizure medication (ASM) that provoked adverse drug reactions (ADRs) in pediatric patients with epilepsy. There was a significant difference among the different causality degrees of ADRs provoked by valproic acid (VPA), levetiracetam (LEV), phenytoin (PHT), oxcarbazepine (OXC), and clonazepam (CZP). Goodness of fit chi-squared test. * *p* value < 0.05. Topiramate (TPM), carbamazepine (CBZ), clobazam (CLB), and gabapentin (GBP).

**Figure 3 children-10-01775-f003:**
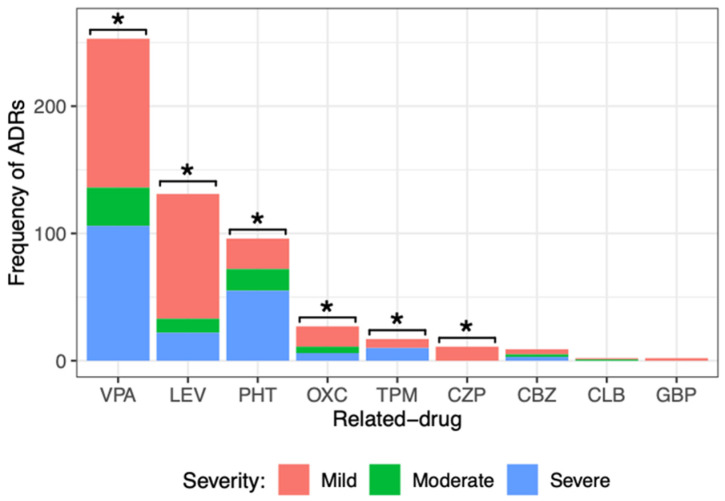
Comparison of the different severity degrees of adverse drug reactions (ADRs) caused by the main ASM administered to pediatric patients with epilepsy. There was a significant difference among the severity degrees of ADRs caused by valproic acid (VPA), levetiracetam (LEV), phenytoin (PHT), oxcarbazepine (OXC), topiramate (TPM), and clonazepam (CZP). Goodness of fit chi-squared test. * *p* value < 0.05. Carbamazepine (CBZ), clobazam (CLB), and gabapentin (GBP).

**Figure 4 children-10-01775-f004:**
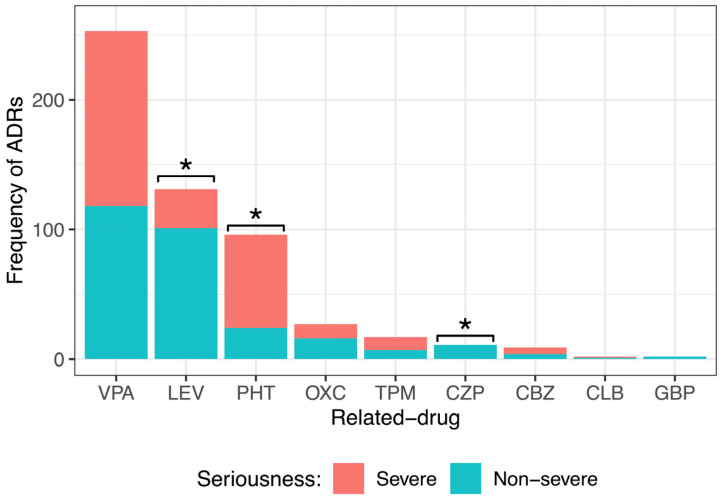
Comparison of the different seriousness degrees of adverse drug reactions (ADRs) caused by the main anti-seizure medication (ASM) administered to pediatric patients with epilepsy. There was a significant difference among the severity degrees of ADRs caused by levetiracetam (LEV), phenytoin (PHT), and clonazepam (CZP). Goodness of fit chi-squared test. * *p* value < 0.05. Valproic acid (VPA), oxcarbazepine (OXC), topiramate (TPM), carbamazepine (CBZ), clobazam (CLB), and gabapentin (GBP).

**Table 1 children-10-01775-t001:** Seizure type, presence of adverse drug reactions due to anti-seizure medication (ASM-ADRs), and electroencephalograms (EEGs) of pediatric patients with epilepsy.

Seizure Type (ILAE 2017)			ASM-ADRs	EEG		
	Total	Present	Absent	Abnormal	Normal	Not Performed
Focal onset	39	30	9	30	2	7
Focal to bilateral tonic—clonic	14	10	4	13	0	1
Focal onset impaired awareness	36	26	10	31	0	5
Focal onset aware	4	4	0	2	1	1
Focal onset motor	25	17	8	20	2	3
Focal motor with impaired awareness	1	1	0	1	0	0
Focal onset nonmotor	3	2	1	1	0	2
Generalized onset	10	7	3	9	0	1
Generalized onset motor	137	96	41	104	6	27
Generalized onset nonmotor	6	2	4	4	0	2
Unclassified	40	16	24	13	6	21
Total	315	211	104	228	17	70

ASM: Anti-seizure medication; ADRs: adverse drug reactions; EEG: electroencephalogram.

**Table 2 children-10-01775-t002:** Univariate and multivariate logistic regression models to analyze factors associated with the presence of adverse drug reactions.

Variable	Adverse Drug Reactions		Univariate Analysis			Multivariate Analysis
Categories	Presence n = 211	Absence n = 104	OR (95% CI)	*p* Value	OR (95% CI)	*p* Value
Sex						
Ref: 2 = Boys	116 (65.9%)	60 (34.1%)	1		1	
1 = Girls	95 (68.3%)	44 (31.7%)	1.12 (0.70, 1.80)	0.648	1.21 (0.72, 2.05)	0.463
Age						
Infant (1 m and <1 yr)	61 (74.4%)	21 (25.6%)	2.90 (1.47, 5.85)	0.002 *	2.80 (1.33, 6.03)	0.007 *
Older infant (1 yr and <2 yr)	28 (65.1%)	15 (34.9%)	1.87 (0.86, 4.17)	0.120	1.91 (0.82, 4.55)	0.137
Preschool (2–4 yr 11 m)	48 (73.8%)	17 (26.2%)	2.82 (1.38, 5.95)	0.005 *	2.69 (1.25, 5.93)	0.013 *
School-aged (5–9 yr 11 m)	40 (70.2%)	17 (29.8%)	2.35 (1.13, 5.01)	0.023 *	2.28 (1.05, 5.10)	0.040 *
Ref: Adolescent (10–18 yr)	34 (50.0%)	34 (50.0%)	1		1	
Seizure type						
Ref: Generalized	105 (68.6%)	48 (31.4%)	1		1	
Focal	90 (73.8%)	32 (26.2%)	1.29 (0.76, 2.19)	0.351	1.29 (0.74, 2.27)	0.378
Unclassified	16 (40.0%)	24 (60.0%)	0.30 (0.15, 0.62)	0.001 *	0.41 (0.19, 0.89)	0.024 *
Therapy						
Ref: Monotherapy	103 (58.5%)	73 (41.5%)	1		1	
Polytherapy	108 (77.7%)	31 (22.3%)	2.47 (1.51, 4.11)	0.001 *	2.08 (1.22, 3.58)	0.007 *
Nutritional status						
1 = Severe malnutrition	72 (61.5%)	45 (38.5%)	0.60 (0.35, 1.03)	0.067	0.58 (0.32, 1.05)	0.073
2 = Mild malnutrition	40 (71.4%)	16 (28.6%)	0.94 (0.47, 1.93)	0.858	0.79 (0.37, 1.74)	0.553
Ref: 3 = Normal weight	88 (72.7%)	33 (27.3%)	1		1	
4 = Obesity	11 (52.4%)	10 (47.6%)	0.41 (0.16, 1.08)	0.066	0.46 (0.16, 1.32)	0.144

Ref: reference category; m: months, yr: years; OR: odds ratio; CI: confidence interval. * *p* value < 0.05.

**Table 3 children-10-01775-t003:** Type, frequency, and distribution by drug of the main adverse drug reactions (ADRs) among pediatric patients with epilepsy.

ADR	VPA	LEV	PHT	OXC	Others *	Total	%
Drowsiness	36	33	6	9	12	96	17.5
Irritability	21	23	1	2	2	49	8.9
Thrombocytopenia	35	4	8	0	2	49	8.9
Alopecia	21	13	1	1	1	37	6.8
Low VPA levels	14	3	12	0	1	30	5.5
Hyperammonemia	24	0	1	0	0	25	4.6
Erythema	9	6	8	2	0	25	4.6
Constipation	11	4	6	1	2	24	4.4
Low PHT levels	3	0	17	0	0	20	3.6
Supratherapeutic VPA levels	13	2	2	0	1	18	3.3
Neutropenia	5	2	2	2	2	13	2.4
Uncontrolled seizure	5	4	1	0	0	10	1.8
Elevated liver enzymes	9	0	1	0	0	10	1.8
Metabolic acidosis	2	0	0	0	7	9	1.6
Rash	2	3	2	2	0	9	1.6
Supratherapeutic PHT levels	2	0	4	1	0	7	1.3
Edema	0	3	2	1	0	6	1.1
Liver damage	2	2	2	0	0	6	1.1
Others	39	29	20	6	11	105	19.2
Total	253	131	96	27	41	548	100%

ADR: adverse drug reactions; VPA: valproic acid; LEV: levetiracetam; PHT: phenytoin; OXC: oxcarbazepine. * Others: topiramate, clonazepam, carbamazepine, gabapentin, and clobazam.

**Table 4 children-10-01775-t004:** Frequency and distribution of the 548 adverse drug reactions (ADRs) that affected organs and systems associated with diverse anti-seizure medication (ASM).

Organs and Systems	VPA	LEV	PHT	OXC	TPM	CZP	CBZ	CLB	GBP	Total	%
Hematologic	113	15	49	5	4	0	3	0	0	189	34.5
Nervous	71	66	14	13	4	10	2	2	1	183	33.4
Dermatologic	41	32	22	8	1	0	0	0	1	105	19.2
Gastrointestinal	25	12	10	1	1	1	4	0	0	54	9.8
Endocrine-metabolic	2	0	0	0	7	0	0	0	0	9	1.6
Cardiovascular	1	1	1	0	0	0	0	0	0	3	0.5
Immunological	0	2	0	0	0	0	0	0	0	2	0.4
Respiratory	0	2	0	0	0	0	0	0	0	2	0.4
Musculoskeletal	0	1	0	0	0	0	0	0	0	1	0.2
Total	253	131	96	27	17	11	9	2	2	548	100%

VPA: valproic acid; LEV: levetiracetam; PHT: phenytoin; OXC: oxcarbazepine; TPM: topiramate; CZP: clonazepam; CBZ: carbamazepine; CLB: clobazam; GBP: gabapentin.

**Table 5 children-10-01775-t005:** Classification by level of severity according to the Hartwig and Siegel Scale of the clinical manifestations of adverse drug reactions (ADRs).

Level	Characteristics	ADR	%
1	No treatment change required	298	54.4
2	Treatment was suspended, but no other medication or antidote was needed, nor was the length of hospital stay increased	87	15.9
3	The treatment was suspended or applied differently, and another medication or antidote was needed, but there was no increase in the length of hospital stay	115	21.0
4.1	The treatment was suspended or applied differently, another medication or antidote was needed, and the length of hospital stay increased by at least one day	11	2.0
4.2	Was the reason for hospital admission	28	5.1
5	Met some of the level 4 conditions and needed intensive care unit admission	0	0
6	Permanent harm was caused	8	1.4
7	Was directly or indirectly related to the death of the patient	1	0.2
	Total	548	100

ADR: Adverse drug reactions.

**Table 6 children-10-01775-t006:** Univariate and multivariate analyses were performed by the generalized estimating equation model to analyze factors associated with the gravity of anti-seizure medication ADRs.

Variable	Adverse Drug Reactions		Univariate Analysis		Multivariate Analysis	
Categories	Severe n = 264	Nonsevere n = 284	OR (95% CI)	*p* Value	OR (95% CI)	*p* Value
Sex						
Ref: 2 = Boys	148 (46.5%)	170 (53.5%)	1		1	
1 = Girls	116 (50.4%)	114 (49.6%)	1.17 (0.75, 1.82)	0.482	0.99 (0.65, 1.51)	0.965
Age						
Infant (1 m and <1 yr)	73 (44.2%)	92 (55.8%)	1.02 (0.53, 1.99)	0.940	0.96 (0.50, 1.85)	0.893
Older infant (1 yr and <2 yr)	30 (49.2%)	31 (50.8%)	1.25 (0.63, 2.48)	0.520	1.03 (0.54, 1.94)	0.939
Preschool (2–4 yr 11 m)	64 (50.0%)	64 (50.0%)	1.29 (0.65, 2.54)	0.450	1.66 (0.88, 3.12)	0.108
Scholar (5–9 yr 11 m)	59 (55.1%)	48 (44.9%)	1.59 (0.72, 3.47)	0.240	1.54 (0.75, 3.18)	0.228
Ref: Adolescent (10–18 yr)	38 (43.7%)	49 (56.3%)	1		1	
Seizure type						
Ref: Generalized	124 (49.0%)	129 (51.0%)	1		1	
Focal	118 (46.5%)	136 (53.5%)	0.90 (0.57, 1.42)	0.650	0.86 (0.56, 1.33)	0.496
Unclassified	22 (53.7%)	19 (46.3%)	1.20 (0.57, 2.56)	0.620	1.46 (0.66, 3.21)	0.337
Therapy						
Ref: Monotherapy	83 (39.0%)	130 (61.0%)	1		1	
Polytherapy	181 (54.0%)	154 (46.0%)	1.84 (1.21, 2.81)	0.004 *	2.11 (1.36,3.30)	0.001 *
Nutritional status						
1 = Severe malnutrition	88 (53.3%)	77 (46.7%)	1.56 (0.99, 2.46)	0.053	1.67 (1.04, 2.67)	0.029 *
2 = Mild malnutrition	49 (57.6%)	36 (42.4%)	1.86 (0.90, 3.82)	0.087	2.06 (1.00, 4.25)	0.046 *
Ref: 3 = Normal weight	110 (42.3%)	150 (57.7%)	1		1	
4 = Obesity	17 (44.7%)	21 (55.3%)	1.10 (0.51, 2.38)	0.797	1.00 (0.57, 1.75)	0.992

OR: Odds ratio; CI: Confidence intervals; Ref: Reference category; m: months, yr: years. * *p* value < 0.05.

**Table 7 children-10-01775-t007:** Association between the preventability of ADRs and the degree of severity presented.

	Preventability			
Severity	Preventable	Probably Preventable	Nonpreventable	Total
Mild	53 (36.8%)	187 (55.0%)	40 (62.5%)	280
Moderate	19 (13.2%)	37 (10.9%)	10 (15.6%)	66
Severe	72 (50.0%)	116 (34.1%)	14 (21.9%)	202
Total	144	340	64	548

**Table 8 children-10-01775-t008:** Association between the preventability of adverse drug reactions and the degree of seriousness presented.

	Preventability			
Gravity	Preventable	Probably Preventable	Nonpreventable	Total
Nonsevere	54 (37.5%)	187 (55.0%)	43 (67.2%)	284
Severe	90 (62.5%)	153 (45%)	21 (32.8%)	264
Total	144	340	64	548

## Data Availability

Not applicable.

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
