# Peer review of "Analysis of Adverse Drug Reactions in Pediatric Patients with Epilepsy: An Intensive Pharmacovigilance Study"

_children, 2023, doi:10.3390/children10111775_

Round 1

Reviewer 1 Report

Comments and Suggestions for Authors

The study aimed to evaluate, by intensive pharmacovigilance over 122 months, the adverse drug reactions provoked by the antiepileptic drugs with the highest prescription rates in the National Institute of Pediatrics in Mexico, which were administered in mono- or polytherapy to Mexican hospitalized pediatric patients with a diagnosis of epilepsy.

The manuscript presents a lot of data, which were properly presented and clearly explained in the text and many tables and figures. The paper also includes an extended discussion of the topic and a limitation section of the study.

Overall, the authors have done a great job, and I commend them for their well-designed and well-presented research! I am looking forward to seeing this paper published. Well done.

Author Response

We thank you very much for your kind comments. We worked hard on this study and really enjoyed the paper!

Reviewer 2 Report

Comments and Suggestions for Authors

Thank you for giving me the opportunity to read and comment a report “Analysis of Adverse Drug Reactions in Pediatric Patients with Epilepsy: An Intensive Pharmacovigilance Study”, by Hernández García E, et al.

The reviewed manuscript evaluates the adverse drug reactions induced by antiepileptic drugs in Mexican hospitalized pediatric patients.

This paper is well written, correctly structured with a suitable research concept and definitely it is of relevance to readers of the journal. However, some suggested minor changes are included in the comments given below:

·         The last sentence of the abstract is a very general conclusion that should not be included in the abstract.

·         All abbreviations used in the tables must be defined in the footnote. I advise authors to review the tables in the manuscript in this regard. For example, in Table 1, the following abbreviations should be defined: AED, ADRs, EEG. In Table 2: OR, CI...

·         In lines 202-204, the authors explain what is meant by an OR result. This is not necessary because the intended audience for this manuscript is familiar with the meaning of OR.

·         The Discussion section usually begins with a brief summary of the main findings. Therefore, the first sentence of the Discussion section could be removed.

·         The authors include as study limitations issues related to the type of eplipesia, other diseases, and concomitant medications. However, the authors do not address the limitations of observational studies.

·         The second part of the conclusion presented by the authors (from line 556) is very general and does not contribute to scientific knowledge. It would be advisable for the authors to modify it and try to respond only to the proposed objectives.

Author Response

  1. Thank you for your kind comments, below we answer your comments and suggestions point by point.

  1. The last sentence of the abstract is a very general conclusion that should not be included in the abstract.

Answer: The sentence was deleted.

  1. All abbreviations used in the tables must be defined in the footnote. I advise authors to review the tables in the manuscript in this regard. For example, in Table 1, the following abbreviations should be defined: AED, ADRs, EEG. In Table 2: OR, CI...

Answer: The abbreviations in the tables were defined in the footnote as you requested.

  1. In lines 202-204, the authors explain what is meant by an OR result. This is not necessary because the intended audience for this manuscript is familiar with the meaning of OR.

Answer: The sentence was deleted.

  1. The Discussion section usually begins with a brief summary of the main findings. Therefore, the first sentence of the Discussion section could be removed.

Answer: We agree with you and several sentences were deleted.

  1. The authors include as study limitations issues related to the type of epilepsies, other diseases, and concomitant medications. However, the authors do not address the limitations of observational studies.

Answer: Now we add a brief explanation of the limitations of observational studies (highlighted in yellow; line 558-564)

  1. The second part of the conclusion presented by the authors (from line 556) is very general and does not contribute to scientific knowledge. It would be advisable for the authors to modify it and try to respond only to the proposed objectives.

Answer: We modify the last part of the conclusions (highlighted in yellow); however, we consider important to mention the conditions of our institute to explain the impact that our study could have.

Reviewer 3 Report

Comments and Suggestions for Authors

Dear authors,

I have read with great interest your manuscript entitled "Analysis of Adverse Drug Reactions in Pediatric Patients with Epilepsy: An Intensive Pharmacovigilance Study". This is a report of a well-designed and properly performed pharmacovigilance study aiming to investigate the adverse drug reaction related to antiseizure medication in a pediatric population.

The results are of great interest to readers and researchers in the field of epileptology (physicians, neurologists, clinical pharmacists, epidemiologists, etc.) so I support their publication. However, some minor concerns have to be addressed.

First, it is impossible to figure out how many patients received a certain drug. For example, how many patients from those 315 included have been treated with valproic acid? Or levetiracetam? So subsection 3.4. should include not only the absolute number of AED-ADRs, but also the number of patients that underwent a specific treatment. I know that most of the patients were under polytherapy, but the absolute number of patients receiving a certain drug should be available.

Second, 

Minor observations:

- Figures would create less white space if compacted.

- Instead of AED, according to the latest recommendations ASM would be more proper.

- The second paragraph of the Conclusion section should be moved to another part of the manuscript, as it does not contain conclusions.

Author Response

Thank you for your kind comments, below we answer your comments and suggestions point by point.

  1. First, it is impossible to figure out how many patients received a certain drug. For example, how many patients from those 315 included have been treated with valproic acid? Or levetiracetam? So, subsection 3.4. should include not only the absolute number of AED-ADRs, but also the number of patients that underwent a specific treatment. I know that most of the patients were under polytherapy, but the absolute number of patients receiving a certain drug should be available.

Answer: We agree that we should give more information about the number of patients receiving the drug. We have omitted since we consider a more exhaustive analysis must be done, related to the number of patients receiving the drug and then analyze the risk that the drug could generate certain AED-ADRs. However, to carry out a formal analysis, we must consider several details that we will explain below.

In section 3.4 we have shown the distribution of the main AED-ADRs by drug, and the table shows the relationship between the type of AED-ADRs with the most frequently drugs suspected of causing that ADRs. In section 3.5 we have shown similar relationship between the organs and systems by drug. From section 3.4 we are just focus on patients having ADRs, i.e. we are not considering those patients who didn’t present AED-ADRs. Then, we are counting the ADRs.

To give the correct information about the number of patients receiving drug we should consider the dynamic of the treatments. Note that the AED-ADRs occurred over time. When patients are included in the research protocol, we collect all the information about the patients and the anti-epileptic treatment at that time (at baseline). We can count the number of patients (considering both patients who had ADRs, and patients who did not) who were taking each drug at the baseline, and it is presented in the following table:

Drug

VPA

LEV

PHT

OXC

TPM

CZP

CBZ

CLB

GBP

Number of children using each drug at baseline

117

187

77

29

17

20

3

13

1

Note that in this table we are omitting information that may be important, for example, the interaction that some of these medications had (we know that the poly-therapy as a higher risk to cause ADRs).

When one drug causes an ADR, it is removed or replaced by another. Which means the number of children using each drug (after the baseline) changes over time. This generates a dynamic in the allocation of medications.

In general, in order to evaluate the risk of treatment, experimental clinical studies (clinical trials) are carried out, where there is control over the assignation of treatment to subjects to guarantee balanced samples, mainly, under the variables sex, age, and perhaps under seizure type, nutritional status, among others. However, to evaluate the risk of treatments in observational studies, it must be considered that the sample under study is not balanced, and therefore statistical methods are used to estimate the causal effects by balancing treatment and control groups on a set of observed covariates, e.g. see Austin and Stuart (2017).

We plan to work in future reports to analyze in deep some of the features listed above, in order quantify the risk of treatment to cause an ADRs and to analyze the dynamic of the recurrence ADRs.

References

Austin P.C., Stuart E.A. (2017). Estimating the effect of treatment on binary outcomes using full matching on the propensity score. Statistical Methods in Medical Research, 26 (6): 2505-2525.

  1. Figures would create less white space if compacted.

Answer: If possible, we modify the figures according to your recommendations.

  1. Instead of AED, according to the latest recommendations ASM would be more proper.

Answer: We change the term AED by ASM

  1. The second paragraph of the Conclusion section should be moved to another part of the manuscript, as it does not contain conclusions.

Answer: We modify the last part of the conclusions (highlighted in yellow); however, we consider important to mention the conditions of our institute to explain the impact that our study could have.